# An exploratory Study of EEG Alpha Oscillation and Pupil Dilation in Hearing-Aid Users During Effortful listening to Continuous Speech

**Tirdad Seifi Ala**[1,2☯*], **Carina Graversen**[1☯], **Dorothea Wendt**[1,3☯], **Emina Alickovic**[1,4☯], **William M. Whitmer**[2☯], **Thomas Lunner**[1☯]

**1** Eriksholm Research Centre, Oticon A/S, Snekkersten, Denmark, **2** Hearing Sciences–Scottish Section, Division of Clinical Neuroscience, University of Nottingham, Glasgow, Scotland, United Kingdom, **3** Department of Health Technology, Technical University of Denmark, Lyngby, Denmark, **4** Department of Electrical Engineering, Linköping University, Linköping, Sweden

☯ These authors contributed equally to this work.
* tirdad.seifiala@nottingham.ac.uk, tial@eriksholm.com

**Data Availability Statement:** There are ethical restrictions on sharing the data set. The consent given by participants at the outset of this study did

## Abstract

Individuals with hearing loss allocate cognitive resources to comprehend noisy speech in everyday life scenarios. Such a scenario could be when they are exposed to ongoing speech and need to sustain their attention for a rather long period of time, which requires listening effort. Two well-established physiological methods that have been found to be sensitive to identify changes in listening effort are pupillometry and electroencephalography (EEG). However, these measurements have been used mainly for momentary, evoked or episodic effort. The aim of this study was to investigate how sustained effort manifests in pupillometry and EEG, using continuous speech with varying signal-to-noise ratio (SNR). Eight hearing-aid users participated in this exploratory study and performed a continuous speech-in-noise task. The speech material consisted of 30-second continuous streams that were presented from loudspeakers to the right and left side of the listener (±30˚ azimuth) in the presence of 4-talker background noise (+180˚ azimuth). The participants were instructed to attend either to the right or left speaker and ignore the other in a randomized order with two different SNR conditions: 0 dB and -5 dB (the difference between the target and the competing talker). The effects of SNR on listening effort were explored objectively using pupillometry and EEG. The results showed larger mean pupil dilation and decreased EEG alpha power in the parietal lobe during the more effortful condition. This study demonstrates that both measures are sensitive to changes in SNR during continuous speech.

## Introduction

Individuals with hearing loss may suffer from a variety of challenges in listening situations such as difficulties in speech perception, which leads to problems with communication and social isolation [1]. In particular, when the listening situation is difficult (e.g., when there is background noise [2]), speech recognition is increasingly more difficult for individuals who

not explicitly detail sharing of the data in any format; this limitation is keeping with EU General Data Protection Regulation, and is imposed by the Research Ethics Committees of the Capital Region of Denmark. Due to this regulation and the way data was collected with low number of participants, it is not possible to fully anonymize the dataset and hence cannot be shared. As a non-author contact point, data requests can be sent to Claus Nielsen, Eriksholm research operations manager at clni@eriksholm.com.

**Funding:** TSA has received funding from the European Union's Horizon 2020 research and innovation programme under the Marie Skłodowska-Curie grant agreement No 765329. WMW was supported by the Medical Research Council [grant number MR/S003576/1]; and the Chief Scientist Office of the Scottish Government. Oticon A/S provided support in the form of salaries for authors CG, DW, EA, TL, but did not have any additional role in the study design, data collection and analysis, decision to publish, or preparation of the manuscript.

**Competing interests:** The authors declare that the research was conducted in the absence of any commercial or financial relationships that could be construed as a potential conflict of interest. The commercial affiliation of authors CG, DW, EA and TL does not alter our adherence to PLOS ONE policies on sharing data and materials.

are hard of hearing [3]. These issues in speech recognition can cause excessive cognitive load, which can in turn lead to negative effects such as difficulties in comprehension [4], recalling the speech [5], [6], fatigue [7] or disengagement from conversations [8]. Hearing devices can assist those with a hearing loss, and may help to reduce some of these limitations by improving memory [9], reducing listening effort [10] and response time [11], as well as providing long-term benefits such as social and emotional improvement [12].

In the literature, behavioral measures, such as the speech reception threshold (SRT), are often used to examine performance in a listening task by normal-hearing and/or hearing-impaired participants [13]. However, this approach may not provide the full picture of the difficulties experienced while listening to speech [14]. Two major issues arise with traditional hearing testing that measures intelligibility in word or short sentence stimuli. The first issue is that in real life, most listening situations involve conversations with free-running, continuous discourse, and do not stop after every few words [15], [16]. The second issue is that even if speech intelligibility is optimal, other cognitive factors might be changing with the difficulty of the task. For example, Sarampalis et al., showed that using a noise reduction scheme in hearing aids did not improve intelligibility but did improve performance in a simultaneous visual task [17]. Houben et al., showed when the speech intelligibility is at ceiling, increasing the signal-to-noise ratio (SNR), reduced the response time of a simultaneous arithmetic task [18]. Both studies concluded that reducing the difficulty of the speech task reduces the cognitive demand which leads to a reduction in listening effort. In this study, we aim to address these two issues by presenting continuous speech, simulating more ecological situations, while objectively monitoring listening effort during different task demands.

Listening effort has been defined as "the deliberate allocation of mental resources to overcome obstacles in goal pursuit when carrying out a [listening] task" [19]. There are myriad ways to assess listening effort [20]: self-report, behavioral responses such as reaction time [17] or by monitoring the changes that occur in the central and autonomic nervous systems during and after speech processing (e.g., [21], [22]). For this latter purpose, two commonly used physiological measures of listening effort are pupillometry, to explore the sympathetic and parasympathetic nervous system activity [23], and electroencephalography (EEG), to measure neural oscillations in the brain [24].

Numerous pupillometry studies have been conducted using different indices, such as peak pupil dilation (PPD) or mean pupil dilation (MPD). They have shown that in more difficult acoustic scenarios, larger PPD and MPD are measures of increased listening effort [25]. For example, in several studies, decreased SNR led to increased PPD or MPD [10], [26], [27]. The pupil dilation has been associated with arousal and resource allocation and is caused by the interaction of the sympathetic and parasympathetic nervous systems.

Studies with similar objectives have also used neuroimaging methods, namely EEG, due to the high temporal resolution it provides. The frontal theta (4–8 Hz) and the parietal alpha (8–13 Hz) are of particular interest. The theta activity in the frontal lobe, has been mostly linked to non-speech processing workload such as pitch discrimination [28], [29], whereas the alpha band has been related to both speech [30] and non-speech [31] related tasks.

Studies utilizing the alpha band, which is usually detected in the posterior regions of the brain, have indicated that these brain oscillations are related to attentional processes in active versus passive listening [32] or different selective attention conditions [33]. However, studies have shown contradictory outcomes with varying listening demand. In some studies, alpha activity increases with more demanding situations [21], [31], [34], whereas in others, alpha activity decreases with more demand [35]–[37]. Some have even reported an "inverted U-shape" form of alpha band which has been associated with listeners "giving up" in increasingly demanding situations, and thus expend no more resources to perform the task [38], [39].

These contradictory results show the ambiguity of interpreting alpha power changes in listening, as listening can involve different cortical processes, depending on the speech material or its presentation [40]. For example, Wöstmann et al., and Deng et al., have shown differences in alpha lateralization when presenting competing speech from contralateral locations [41], [42].

The aforementioned studies on listening effort, both in pupillometry and EEG, were conducted using stimuli consisting of mostly single words, tones or short sentences. However, there is a need for studies in more ecological situations, to match those experienced by hearing-impaired individuals in everyday life. To begin investigating physiological changes during a listening task in more ecologically valid situations, we conducted an exploratory study where continuous auditory news clips were presented to hearing-impaired participants at two different SNRs. This enabled us to explore changes in pupillometry (MPD) and EEG (theta and alpha power) with SNR that extend the knowledge about the physiological changes of listening effort in continuous speech.

While both pupil dilation and EEG alpha power have been widely used for detecting changes in listening effort, they have not been reported to correlate to one other during tasks involving short duration stimuli [36]. The lack of correlation in these measures might be due to the slow response of pupil dilation compared to the fast changes in EEG. For this reason, presenting longer stimuli in this study will also provide the chance to look for a delayed correlation between the two measurements.

## Methods

### Participants

Eight native Danish-speaking test adults (2 females) with an average age of 70 ± 12 years participated in the study and signed a written consent form prior to study onset. Ethical approval for the study was obtained from the Research Ethics Committees of the Capital Region of Denmark. All test participants were experienced hearing-aid users with symmetrical, mild, sensorineural hearing loss. The pure-tone average of air conduction thresholds at 0.5, 1, 2 and 4 kHz was 31 ± 5.5 dB HL. The average difference between the left and right ear in air conduction hearing thresholds for 0.5, 1, 2 and 4 kHz was a maximum of 5 dB.

The participants were fitted binaurally with behind-the-ear Oticon Opn1 mRITE hearing aids with miniFit Speaker Unit 85. Domes used in the test corresponded to what the test subject was currently using: either miniFit open domes or miniFit Bass domes with 1.4 mm vent effect. Noise reduction and directional microphones were deactivated so that the hearing aids just provided individualized audibility via the proprietary gain and frequency prescription rule. Volume control and the mute function were also deactivated to prevent the test subjects from changing the gain during testing.

### Apparatus

The experiment was set up in a double-walled sound-proof booth. The experimental setup consisted of three loudspeakers positioned at ±30˚ and +180˚ azimuth relative to the participants. The loudspeakers in the front hemifield were the target and contralateral distractor locations, symmetrically off-center to counterbalance any asymmetrical hearing abilities, and the loudspeaker in the rear hemifield presented 4-talker babble noise to increase task complexity. The eye tracker and a computer screen for displaying the instructions and the questions were positioned in front of the participants in a way not to cause acoustic shadowing. The spatial setup of the test is illustrated in Fig 1A.

Stimuli were routed through a sound card (RME Hammerfall DSB multiface II, Audio AG, Haimhausen, Germany) and were played via loudspeakers Genelec 8040A (Genelec Oy,

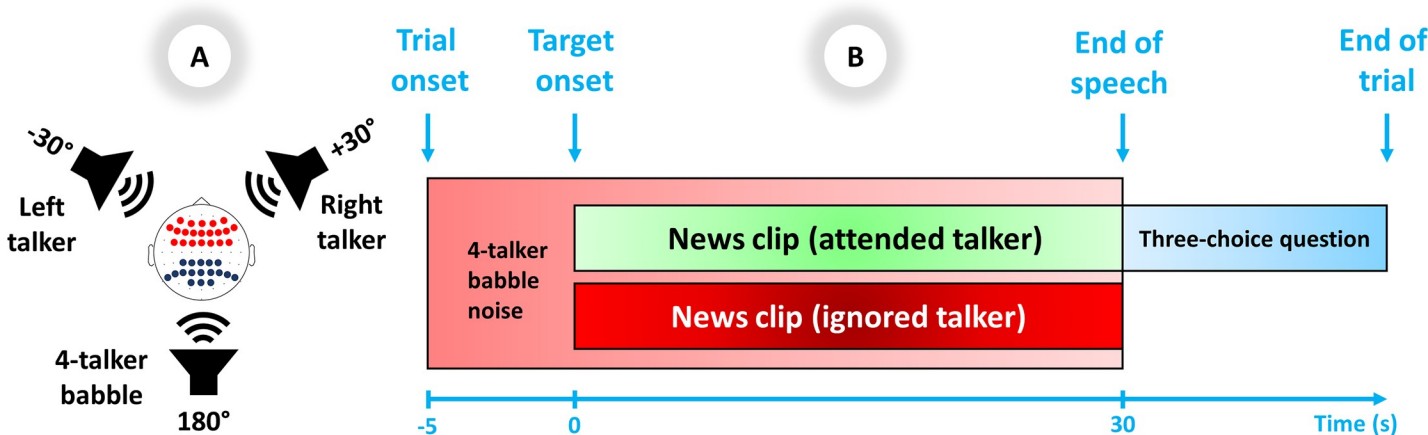

**Fig 1. A)** Spatial setup of the experiment: Test subjects attended to target stimuli from a front loudspeaker ±30˚ to the left or right. The contralateral front loudspeaker presented the talker to be ignored. The rear loudspeaker presented 4-talker babble. In the superior view of the head, EEG electrode locations for frontal theta are shown in red dots and parietal alpha are shown in dark blue dots. **B)** Trial scheme: The target and distractor speech were presented 5 seconds after the onset of the 4-talker babble and then continued for 30 seconds, followed by a three-choice question regarding the content of the attended target audio clip.

Iisalmi, Finland). Pupillometry and EEG devices were used to collect the physiological data. Pupil diameters of the left and right eyes were recorded by an SMI iView (SensoMotoric Instruments, Teltow, Germany), RED250 mobile system with a sampling frequency of 60 Hz. EEG data were recorded by a BioSemi ActiveTwo amplifier system (Biosemi, Netherlands) with a standard cap including 64 surface electrodes mounted according to the international 10–20 system with a sampling frequency of 1024 Hz. The cap included DRL and CMS electrodes as references for all other recording electrodes. All electrodes were mounted by applying conductive gel to obtain stable and below 50 mV offset voltage.

## Stimuli

Non-dramatic Danish news clips of neutral contents were used for the target and contralateral distractor speech (30 seconds), while the 4-talker babble noise (35 seconds) was provided by Danish audiobooks. The target and distractor speech were read by a randomized male or a female speaker, and for each trial the target and distractor were never the same gender.

The A-weighted sound pressure level at the center of the room was 50 dB for the babble and 65 dB for the target on every trial. The contralateral distractor level was either 65 dB or 70 dB on each trial to generate two different SNR conditions: 0 and -5 dB. For this study, SNR was defined as the long-term average sound level of the target signal (with pauses longer than 200ms being cut out) compared to the competing front talker only. Although both SNRs were relatively low compared to common environments for hearing-aid wearers (cf. [43]), we will refer to the 0 dB and -5 dB SNR conditions as "high SNR" and "low SNR", respectively.

## Procedure

There were 54 trials for each SNR, randomly distributed across all 108 trials. Each trial (Fig 1B) consisted of 35 seconds of 4-talker babble played in the background. The target and distractor speech were presented 5 seconds after the onset of the babble (i.e., after the baseline period) and then continued for 30 seconds, followed by a three-choice question regarding the content of the attended target audio clip [e.g., "Who warns against the dangers of discrimination?" (English translation)]. Participants were given a rest period every 36 trials, while minor breaks were given between every 8th trial.

Before each trial, the participants were instructed on the screen to pay attention to the target on the right or left side and ignore the talker on the other side and the babble behind them. The location of the target (i.e., right or left front loudspeaker) was also randomized between each trial.

## Behavioral measurement

To motivate the participants to maintain their attention to the target speaker, a three-choice question was displayed on the screen immediately after each trial, which they were instructed to answer. The percentage of correct answers per SNR was registered to reflect both hearing-in-noise abilities and attention to the target.

## Pupillometry measurement

To analyze the pupillometry data, eye blinks were first detected as pupil diameter data with values two standard deviations (SD) below the trace's mean value and then removed. Missing gaps caused by blink removal were linearly interpolated 80 ms before and 150 ms after the blinks to match the rest of the trace. Other high frequency artifacts potentially caused by unrelated physiological processes were also removed from the signal by means of a moving average filter with a symmetric rectangular window of 600 ms length. Eventually, only trials with more than 75% reliable data were kept for further analysis. No subject had less than 80% good trials, so all of them were kept for further analyses.

As a normalization method, subtraction of the pupil baseline value (-4 to 0 sec., with 0 indicating the onset of the target) was used to extract task-related pupil activity. Data were averaged across the tested conditions (high and low SNR), and each data point within 5-second intervals was averaged together (e.g., 0–5 sec., 5–10 sec. and so on). This provided an opportunity to compare the results of pupillometry with EEG power spectral analysis in the same time intervals. MPD was applied since it is more robust compared to PPD in longer stimuli designs, as MPD extracts all the information within 30 seconds of data. In contrary, PPD usually happens only in the first few seconds of the target onset and gives no further information for the rest of the stimuli.

The longer stimuli also provide the opportunity for exploring other features within pupil data such as the difference in the MPD. For this reason, the difference of time-windowed mean pupil dilations was compared between low vs. high SNR.

## EEG analysis

The EEG trials were segmented from -5 to 32 seconds after stimuli onset (the last 2 seconds only included to avoid edge effects of the spectral perturbation). First, 50 Hz power line noise was rejected with a notch filter with quality factor of 25. Then, a 3rd-order zero-phase Butterworth bandpass filter with a cutoff frequency of 1 to 40 Hz was applied to the data, which was afterwards down-sampled to 256 Hz. Bad channels were automatically detected if they had values higher than three SD in more than 25% of the whole recording. A maximum of 3 out of 64 channels were detected as bad across all participants, in which case, the data were interpolated using spline interpolation in the EEGLAB toolbox [44].

**EEG denoise.** To remove artifacts in EEG data, joint decorrelation [45], which is an improved method over denoising source separation (DSS), was applied. The bias filter in this method for denoising was chosen as the average of trials. Such a bias filter enhances the optimal weights for independent components in a way that components have the most repeatability across all trials. Each of the extracted components were ranked according to the power of their mean divided by the total power, which implies that the first component has the strongest

possible mean effect relative to overall variability and hence has the highest chance to be neural activity related. To decide how many of the components should be kept and then backpropagated to the sensor level, a surrogate procedure took place. If the score of the component was higher than the 95% confidence interval of the surrogate data, the component was regarded as neural activity; otherwise, it was discarded [45]. It should also be noted that no trial was discarded due to poor signal quality and the results are based on the average of all recorded trials.

**Event-related spectral perturbation.** To assess how the EEG power spectra changed compared to the baseline, the event-related spectral perturbation (ERSP) method was used. The main characteristic of this method is that the EEG power over time within a predefined frequency band is displayed relative to the power of the same EEG derivations recorded during the baseline period [46]. The formula to calculate the ERSP is as follows:

$$ERSP_t(\%) = \frac{A_t - R}{R} \times 100$$

where $A_t$ is the absolute power of the post-stimulus signal in time window $t$ and $R$ is the absolute power of the baseline signal (-4 to 0 sec.) in a specific frequency band. ERSP for both theta (4–8 Hz) and alpha (8–13 Hz) bands were calculated separately using the Welch method [47]. For each trial, the ERSP was calculated for each 5-second interval (e.g., 0–5 sec., 5–10 sec. and so on). The average over all trials for each interval was calculated for each condition. To obtain a more robust estimate of the changes in frontal theta, the ERSPs of electrodes AF3, AF4, AF7, AF8, AFz, F1, F2, F3, F4, F5, F6, F7, F8, Fz, FC1, FC2, FC3, FC4, FC5, FC6, and FCz (shown in red dots in Fig 1A) and for changes in parietal alpha the ERSPs of electrodes CP1, CP2, CP3, CP4, CPz, P1, P2, P3, P4, P5, P6, P7, P8, Pz, PO3, PO4, POz (shown in dark blue dots in Fig 1A) were averaged together.

**Alpha lateralization.** Alpha lateralization was also investigated to see if attending right vs. left stimuli elicits different responses in different hemispheres. To do so, the alpha power when the participants were attending to the right target was subtracted from the alpha power when the participants were attending to the left target for both lateral hemispheres. Then, right hemisphere was compared to left hemisphere to see if they respond differently, depending on the location of the target.

## Statistics

For statistical evaluations, IBM SPSS Statistics v.24 was used. First, the normality assumption of data was checked numerically by Kolmogorov-Smirnov's test and visually by Q-Q plot [48]. The comparison of the *performance* results based on *SNR* was undertaken using paired t-test. For *MPD*, *difference in MPD*, *theta* power, *alpha* power repeated measure ANCOVA was used, with *SNR* as the predictor and *Time* (0–5, 5–10, 10–15, 15–20, 20–25, 25–30 sec.) as the covariant factor. The main effect of *SNR* and the interaction effect of *SNR* and *Time* were investigated. For alpha power lateralization, the same *Time* windows as covariant factors were used, but unlike other dependent variables, right vs. left *hemispheres* were used as predictors of the repeated measure test. Additionally, partial correlation was also performed on difference of *MPD* and *alpha* power (High SNR–Low SNR), with *Time* being the covariant factor. P-values of less than 0.05 are considered as significant differences.

## Results

In this section, the results of behavioral responses, MPD, parietal alpha and frontal theta power for the two test conditions (high and low SNR) will be presented. First the normality assumptions were confirmed for each measurement with Kolmogorov-Smirnov's test and Q-Q plots.

## Behavioral results

The mean correct percentage was significantly higher [t(7) = 5.56, p = 0.001] in the high SNR condition (76.7%) than the low SNR condition (61.8%), reflecting that the participants benefited from higher SNR in terms of understanding the contents of the speech. Also, the above chance performance for the low SNR suggests that the speech in the worst condition was still partly intelligible.

## Listening effort

To estimate the listening effort during the task, pupillometry and EEG were used as measures of the effect of task demand induced by different SNRs. The results indicated significantly larger MPD [F(1,46) = 18.65, p < 0.001] in the low SNR compared to the high SNR. No interaction between SNR and Time were found [F(1,46) = 0.044, p = 0.836]. The normalized MPD graph of averaged trials over the 30-second period is shown in the left panel of Fig 2C, and the averaged MPD over the range of 30 seconds of stimuli for each individual participant are shown in the right panel of Fig 2C. The comparison between the difference in MPD showed no significant change between low vs. high SNR [F(1,46) = 2.685, p = 0.108] nor an interaction between SNR and Time [F(1,46) = 2.132, p = 0.151]. These results show that in longer stimuli mean pupil is still more sensitive to task demand than its changes.

The alpha ERSP in the parietal lobe showed less activation [F(1,46) = 4.63, p = 0.037] in the low SNR compared to the high SNR. Similar to pupillometry results, no significant interaction between SNR and Time was observed [F(1,46) = 0.016, p = 0.899]. Fig 2A shows the brain topographical maps and activated regions in the alpha band. The left panel of Fig 2B shows the parietal alpha ERSP graph of averaged trials over the 30-second period and the right panel of Fig 2B illustrates the averaged parietal alpha over the range of 30 seconds of stimuli for each individual participant.

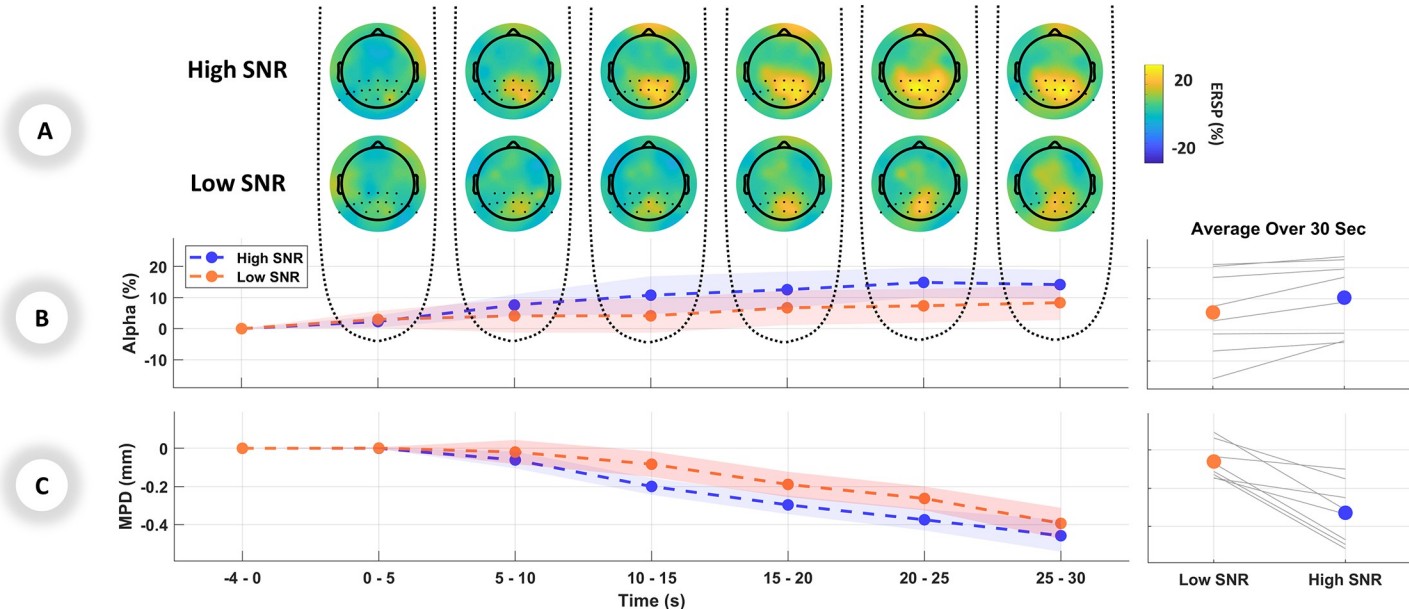

**Fig 2.** Listening effort indicated by physiological measurements: **A)** Grand average EEG topographical maps in windows of 5 seconds during presentation of stimuli. The first and second rows show the topographical maps for high SNR and low SNR respectively. **B)** Left panel: ERSP changes in percentage for the alpha band over the parietal region, averaged over each 5-second period. Right panel: Individual and mean average of parietal alpha over 30 seconds **C)** Left panel: MPD changes in millimeters for the pupillometry data, averaged over each 5-second period. Right panel: Individual and mean average of MPD over 30 seconds. Standard errors are shown as shaded area in **B** and **C**.

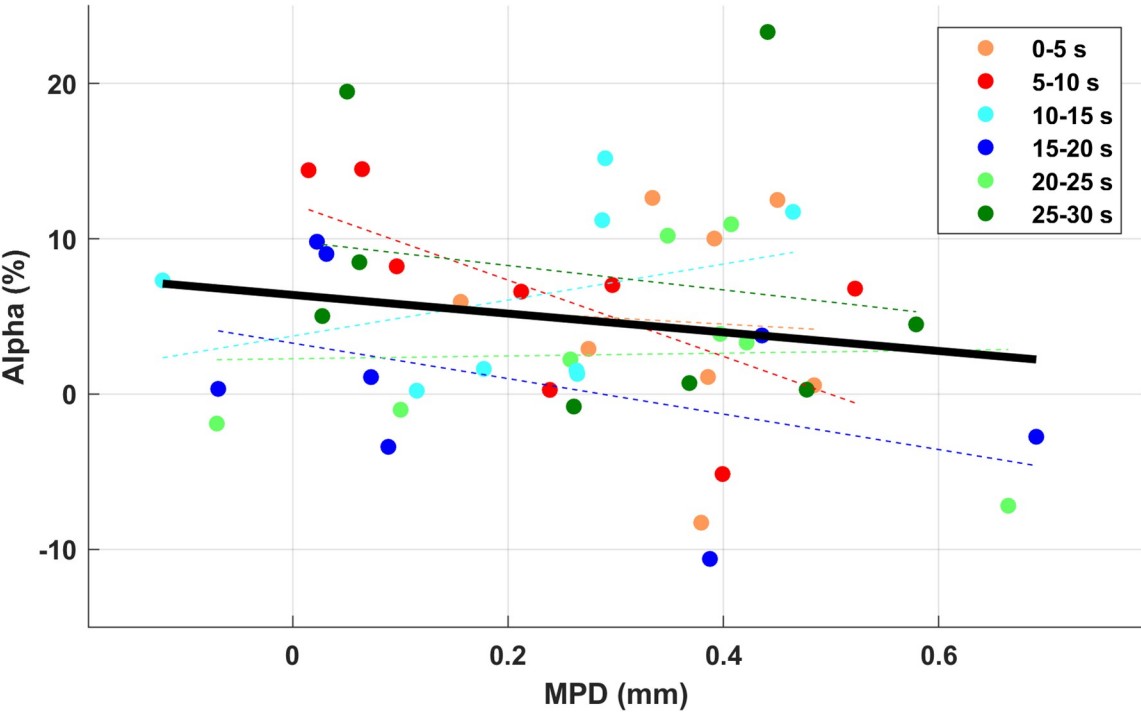

**Fig 3. The partial correlation between the difference of MPD and alpha power, with time as the covariance.** No significant correlation was found.

Investigation of the frontal theta did not show any significant effect of SNR [F(1,46) = 0.860, p = 0.358], nor the interaction between SNR and Time [F(1,46) < 0.01, p = 0.984].

The partial correlation between the difference of MPD and alpha power with Time factor as covariance did not show any significant result [r(45) = -0.168, p = 0.258] (Fig 3). Given the expected delay between EEG and pupil response, the relationship between alpha power in each 5-s time window and MPD in each subsequent time window was also compared; there was, however, no significant correlation.

## Alpha lateralization

For alpha lateralization, the difference between "attended right" and "attended left" in the right and left hemispheres did not show any significant difference [F(1,46) = 0.049, p = 0.826], suggesting the location of the target did not elicit different responses between hemispheres.

## Discussion

The aim of this exploratory study was to demonstrate how listening effort in hearing-impaired participants can be affected by different SNR conditions during continuous speech. The speech material used in this study was not typical short sentences; instead, it was comparatively longer, connected speech in fixed SNR conditions. This design was chosen to obtain a more ecologically valid approach, since communication in everyday life often includes listening and being exposed to longer stimuli rather than just a few words or single sentences. For this purpose, the designed protocol consisted of 30-second news clips with high (0 dB) and low (-5 dB) SNRs. Participants were hearing-aid users, who were instructed to focus on one talker while ignoring the other. Pupillometry and EEG were used to reveal changes in the nervous system reflecting listening effort during 30 seconds of the speech presentation.

## Pupillometry

Many studies have shown that the pupil response is sensitive to changes in listening effort during presentation of short stimuli; a larger dilation relative to the baseline has been shown with increased listening effort [25], [49], [50]. Using longer stimuli (30 seconds) in this study, however, resulted in smaller dilation relative to the baseline, in both low and high SNR conditions (Fig 2C). This is probably caused by the sensitivity of the pupil to task alertness, which could be more pronounced in the first seconds of the trial and has previously been observed in longer pupil data collection as well (e.g., [37], [51]). The relative decrease of MPD measurement over 30 seconds might be due to evoked pupil dilation to the background noise in the baseline. Nevertheless, it is clear from Fig 2C that in the harder condition, MPD was still higher (less negative) for continuous speech, which demonstrates increased listening effort for sustaining attention. Larger pupil dilation during demanding conditions has been associated with increased workload and a greater allocation of resources to perform the listening task [27].

In addition to pupil changes during listening, studies (e.g., [52], [53]) have shown pupil size can also change after the listening phase and during retention. Sustained increase of pupil dilation in the retention phase can happen in more demanding conditions. While in this study there is no retention phase, it can be argued that presenting long, continuous-speech stimuli requires gradual retention, especially towards the later parts of the stimulus, which can affect pupil dilation.

## Alpha power

Using alpha power of the EEG signal as an outcome measure for listening effort has resulted in contradictory results in previous studies. While some studies suggest the relationship between alpha power and task difficulty is direct, i.e. more difficulty equals increased alpha (e.g., [21], [31], [38], [54], [55]), others have shown the inverse, i.e. more difficulty equals decreased alpha (e.g., [36], [37]). For example, Petersen et al., showed that during recognition of monosyllabic digits, greater power in the alpha band was observed with increasing severity of hearing loss and increasing use of working memory (before the task became too difficult) [38]. On the other hand, Miles et al., reported that in a speech recognition task, parietal alpha was decreased during a demanding situation when the spectral content of the signal using noise vocoding and speech intelligibility were changed [36]. Though clearly a disputed concept, listening effort related changes in alpha power are probably a function of the speech material used and may vary based on the definition of the "listening" task and/or different demands which require top-down or bottom-up processing.

During the continuous discourse in this study, alpha band in the parietal lobe was lower in magnitude in the demonstrably harder condition (low SNR) compared to the easy condition (high SNR) (Fig 2B). It is not clear which underlying mechanisms drive the activation or suppression of alpha power in demanding situations. Two common and conflicting theories on alpha power in effortful situations exist. One theory explains that increased alpha is a sign of suppression of unattended sound sources [56] and inhibition of task-irrelevant cortical regions [57], which as a consequence should increase alpha with increased difficulty. On the other hand, "cortical idling" theory states that synchronized (i.e. increased) alpha is a correlate of a deactivated cortical network [58] which then facilitates better performance [59].

Our results are in-line with the second theory: in the low demand situation alpha power and performance increase, which suggest an indirect and inverse relationship between alpha and effort i.e. decreased alpha equals more effort. The conflict of our results with some of the other literature might be explained by one study by Jensen et al., in which participants performed the Stenberg task to see how parietal alpha alters with higher workloads [60]. They

observed increased alpha power with increased workload which conflicted with the results from other working memory tasks, namely n-back task, in which decreased alpha activity was observed with higher demand [61]. They concluded that in the Stenberg task the brain response is different when the encoding and retention phases are temporally independent from each other, compared to an n-back task where these phases are overlapping and require a constant update of information in the working memory. Given the nature of the stimuli in the current study, sustained attention and constant updating of working memory is required over 30 seconds of speech presentation. The entangled encoding and retention phases might call for decreased alpha activation when it is more difficult. This notion goes along with other studies that showed optimal sustained attention performance is linked to greater alpha oscillation [37], [62] and thus can be interpreted as inversely related to listening effort.

The spatial setup with a contralateral distractor in this study provided the chance to look at the alpha lateralization in a more realistic situation with background noise. However, unlike previous studies [41], [42], no difference between hemispheres was observed in the data. One key difference between those previous studies and the current study is the addition here of four-talker babble noise at 50 dB from directly behind the listener. The presence and/or location of the background noise in the current study may have obscured any indication of alpha lateralization. Another difference between the current and previous studies is that listeners in the current study were bilaterally aided, which may have also affected alpha lateralization. Further studies are required to fully explore this lack of alpha lateralization, but this result highlights the potential importance of using a background noise in spatial attention tasks.

### Pupil dilation and alpha power correlation

The co-registration between pupillometry and EEG has also been shown in previous studies, such as [34] and [36], who used vocoded short sentences and 4-talker babble background noise. They observed an increase in pupil dilation and a decrease in alpha power in the more spectrally degraded 6-channel speech, as compared to the 16-channel condition, but found no correlation between them. In-line with those results, the current study showed no (negative) correlation between MPD and alpha power (Fig 3), despite the high consistency between the two modalities i.e. increased MPD and decreased alpha. This lack of correlation could speak for different cognitive functions presented by each of them. After all, the driving mechanisms for pupil dilation and alpha power originate from different areas in the nervous system. Pupil diameter is suggested to reflect different neuro-modulatory systems such as locus coeruleus–noradrenergic (LC-NE) which increases task-relevant neuronal gain in cerebral cortex in rapid dilations [63] or basal forebrain which modulates the state of cortical activity during sustained activity [64]. On the other hand, the posterior supramarginal gyrus (SMG) and temporoparietal junction (TPJ) are mainly responsible for generating alpha activity during effortful listening [21], [30]. This suggests that even though both measurements have been widely used for assessing listening effort, they might be generated independently and capture different cognitive aspects.

### Theta power

The theta band, which oscillates in slower frequencies than alpha band, has been widely recognized as neural correlates of "cognitive effort" in many non-auditory working memory tasks [65–67]. However, hearing studies show that the modulation of theta band mainly happens during non-speech tasks. For example, when the participants were asked to recognize the highest pitch when exposed to square waves, the frontal theta showed an increase in more demanding situation where retention was required to perform the task [29]. On the other hand in a

speech-related task, [35]demonstrated that degrading the SNR in a linguistic task consisted of disyllabic words in children with asymmetric sensorineural hearing loss did not result in higher frontal theta activation.

The current study, in which the task heavily relies on linguistic contents, showed no changes of theta band due to changes in SNR. One area that might be intriguing for future studies would be to look for the role of theta activation in speech vs. non-speech related tasks, as it seems the reports on "effortful" theta are mainly based on non-linguistic contents such as pitch discrimination. However, this should not be misinterpreted that theta band does not play a role in linguistic processing. Many studies have shown that by decoding the low-frequency cortical responses (mainly EEG theta band) with the speech envelope, a classifier can be formed to discriminate between attending two competing talkers at the same time [68], [69].

## Limitations and summary

There are several limitations in this study. The first is that two factors interplay to determine the use of mental resources during listening effort. One factor is task-related, which depends on the difficulty of the task [70], and the other factor is individual-related, which varies with motivation [19]. The aim of the current study was to manipulate task demand by change in SNR, and not motivational factors, to vary listening effort. It cannot be ruled out, though, that individual-related effects also played a part in shaping listening effort. The second limitation is the low number of participants ($n = 8$) recruited for this experiment. Although this affects the statistical validation, the normality assumption of the data was checked by both Kolmogorov-Smirnov and Q-Q plot. Also, as an initial investigation into these physiological measures in a continuous task, we aimed to rely less on interpreting the p-values and more on the high consistency of individual responses by providing single-subject results (right panels in Fig 2).

In summary, this study provides an initial demonstration that pupillometry and EEG can be applied as indices of task-related listening effort during long speech segments in hearing-impaired participants. Both modalities confirmed increased effort with decreasing SNR in the continuous auditory stimuli. These results could be viewed as initial steps towards using objective measurement of listening effort in more ecologically valid situations, which is currently lacking in the hearing science. As there was no correlation between the two measurements, it remains to be seen which factors can systematically alter both in a continuous discourse paradigm. This would help elucidate their cognitive roles in sustained attention and how they lead to listening effort.

## Conclusion

In this exploratory study pupillometry and EEG were used to assess aspects of listening effort of hearing-aid users in a continuous speech setting. When listening to 30-second news clips, presented from either a right or left target in the presence of 4-talker babble noise, higher listening effort was observed with both pupillometry (larger mean pupil dilation) and EEG (less parietal alpha power) for the more demanding and effortful condition (lower SNR).

## Acknowledgments

The authors would like to thank Renskje K. Hietkamp, Patrycja Książek and Eline Borch Petersen for their contribution in preparing the experiment and data collection. We would also like to express our gratitude to Gitte Keidser and Lorenz Fiedler for fruitful discussions of the paper.

## Author Contributions

**Data curation:** Carina Graversen, Dorothea Wendt.

**Formal analysis:** Tirdad Seifi Ala, Dorothea Wendt.

**Methodology:** Tirdad Seifi Ala, Emina Alickovic.

**Supervision:** Carina Graversen, Dorothea Wendt, William M. Whitmer, Thomas Lunner.

**Validation:** Carina Graversen, Dorothea Wendt.

**Visualization:** Tirdad Seifi Ala.

**Writing – original draft:** Tirdad Seifi Ala.

**Writing – review & editing:** Carina Graversen, Dorothea Wendt, Emina Alickovic, William M. Whitmer, Thomas Lunner.

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
