## [Decision Letter · Decision Letter 0]

27 Mar 2020

PONE-D-20-04937

Assessment of EEG Alpha Oscillation and Pupil Dilation During Continuous Speech in Hearing-Aid Users; A Measure of Listening Effort?

PLOS ONE

Dear Dr. Tirdad

Thank you for submitting your manuscript to PLOS ONE. After careful consideration, we feel that it has merit but does not fully meet PLOS ONE’s publication criteria as it currently stands. Therefore, we invite you to submit a revised version of the manuscript that addresses the points raised during the review process. Two reviewers have reviewed the manuscript and advise amendments to the description of the methodology and interpretation of the results.

We would appreciate receiving your revised manuscript within 45 days of this notice. To enhance the reproducibility of your results, we recommend that if applicable you deposit your laboratory protocols in protocols.io, where a protocol can be assigned its own identifier (DOI) such that it can be cited independently in the future. For instructions see: http://journals.plos.org/plosone/s/submission-guidelines#loc-laboratory-protocols

We look forward to receiving your revised manuscript.

Kind regards,

Ifat Yasin

Academic Editor

PLOS ONE

Journal Requirements:

We note that one or more of the authors are employed by a commercial company: Eriksholm Research Centre, Oticon A/S

Reviewers' comments:

Reviewer's Responses to Questions

**Comments to the Author**

1. Is the manuscript technically sound, and do the data support the conclusions?

Reviewer #1: No

Reviewer #2: Yes

2. Has the statistical analysis been performed appropriately and rigorously? 

Reviewer #1: I Don't Know

Reviewer #2: Yes

3. Have the authors made all data underlying the findings in their manuscript fully available?

Reviewer #1: No

Reviewer #2: No

4. Is the manuscript presented in an intelligible fashion and written in standard English?

Reviewer #1: Yes

Reviewer #2: Yes

5. Review Comments to the Author

Reviewer #1: This paper describes a study where listening effort was measured in response to continuous speech passages. The major novelty of the study is that the stimuli are longer than what is typically used in most studies of effort, which usually use single sentences. The current experiment used listeners with hearing loss who were fit with hearing aids. The data suggest that pupil dilation declines over the course of prolonged listening, but declines at a slower rate (i.e. remains more dilated) when the speech is more difficult. The EEG results were less clear, and there was no correlation between the two measurements on a global level.

The study offers a potentially valuable contribution to the literature on listening effort. The major flaw – the number of participants is addressable without changing the design of the study. The other limitations of the manuscript itself are also addressable. I will outline the major points below, as well as some minor things that could improve the paper. At the moment, I do not think the paper is worth reviewing again without additional data collection from a larger number of participants.

Major comments:

1) Number of participants

The number of participants in the study is too small. The authors attempt to soften this limitation by explaining in line 376 that individual data are rather consistent in the pattern across conditions. This is mostly true for the pupil data and not for the EEG data. Regardless, it appears to me as if the study is simply not complete. Why stop at 8 participants? As the main motivation for the study appears to be the establishment of the method of measuring responses to continuous speech rather than the use of listeners with hearing aids, it would seem feasible to add power to the study by having more easily accessible sample of listeners with typical hearing. If the goal is to learn something about hearing impairment/the use of hearing aids, there is just too small a number of listeners.

2) Conclusion: “This study demonstrates that pupillometry and EEG can be effectively used to assess listening effort”

This seems premature to me. I don’t see how the authors justify calling this method “effective”, or how they have established that the changes in EEG/pupil size index the same thing. I would recommend rewriting the conclusions to convey only what is clearly indicated by the data, and leave speculation in the discussion section. As you might guess, I think that part of why this conclusion is premature is because the number of participants is too small. To validate this new approach to measuring effort for continuous speech, you really ought to make sure you understand the extent of individual variability and establish that it isn’t just something that explains these eight people.

3) Methods – please explain the physical location of the maskers - why was there a masker 60 degrees lateral of the target and also one masker behind the listener? What was the rationale for having a slightly-left or slightly-right location of the talker, and for the target location to randomly alternate between trials? For a study that emphasizes the need to incorporate ecological validity, this seems like an odd setup that is perhaps contrived to exploit specific kinds of hearing aid processing.

4) Review of literature

The review of literature should actually inform the reviewer of how some of the critical knowledge was acquired rather than just make statements followed by names and years. In general, there are very many places in the paper where the literature review is not helpful, because it either 1-makes statements without any explanation of how the knowledge was acquired, or 2-conflates a hypothesis with fact. I will give some examples:

Line 48-49: “Even if individuals can hear what is being said, they may need to put more effort in to process the auditory input (Lunner et al., 2016).”

Tell us how this is known. As it is written, you’re just asking the reader to trust Lunner that this is true. How was this conclusion established? There are lots and lots of published papers that have established this kind of conclusion using various methods. It would be an incomplete paper without actually describing the work that has come before. We can’t just make statements and attach names to them as if the name alone provides the authority to trust the claim. Tell us that “Author A did a study where B was measured using C as a stimulus. Group D performed differently than group E, allowing us to conclude F.” This is much more informative than “F is true (Author A).”

Line 43-44: “free-running, connected discourse, that mainly exists in real life situations, can hardly be analyzed by word or sentence intelligibility (Speaks et al., 1972)”

What is meant by this? Why can’t it be analyzed that way? Tell the reader why this is the case. If you delete the sentence that begins with “Henceforth”, then it could be clearer.

Line 355: “…the frontal theta showed an increase in more demanding situations”

What is meant by “more demanding situations”? This is another one of many examples in this paper where the literature is not reviewed in a way that allows the reader to understand what work was actually done. Same for the following line where it is not clear what is meant by “a linguistic task”.

Line 36: “helping to reduce these limitations in everyday life” – do these studies actually demonstrate that hearing aids lead to reduced fatigue, increased social participation and engagement in conversations and improve recall? I suspect that these things have not actually been established, and it is important to recognize what is “hypothesis” and what has actually been empirically demonstrated.

Line 34: “recalling the speech” – is the Rönnberg study an empirical study demonstrating this, or a theory paper? Studies on recognition memory/recall have been done by Smiljanic, Van Engen, Gilbert, and others.

Starting at line 300: the discussion of studies by Peterson and by Miles is not as clear as it could be. The specific stimulus that these studies used is very important. The manuscript says that Peterson used monosyllabic digits – this is helpful because it means that there was almost no language processing like syntactic or semantic context processing. No mention was made of the stimuli used by Miles, which makes this section incomplete. Is it possible that the difference between these contradictory studies simply reflects the different demands of the stimuli used?

Line 325-333: the discussion of the retention interval in EEG studies is good, but it highlights the absence of a similar discussion of retention interval in pupillometry studies, e.g. by Piquado et al., and by Winn & Moore.

5) Figure 2 – the design is very nice, and easy to understand. But there are two things to change: first, do not use a red & green color pattern, because a common form of colorblindness means that readers cannot distinguish these colors. Any other color pair would be better. This goes for both the EEG data, pupil data, and the brain topomaps. Second, please produce this figure with higher resolution, as it appears very blurry.

6) The title of the paper is entirely uninformative. The title should indicate something about what was found. As it is currently written, it is more like “click bait”

MINOR COMMENTS

Line 45: improper use of the word “henceforth” - that would mean “from this point forward…”

Abstract: the sentence “This sustained attention requires cognitive resources, the expenditure of which leads to listening effort.” Could be removed, as it is reasonable to rephrase as “X requires X, leading to X”, where X is the same concept repeated multiple times redundantly, just with different words.

The abstract is misleading when it implies that pupillometry is a well established method to look at sustained attention or effort. It is in fact a well established way of looking at momentary, evoked or episodic effort, not sustained effort. But I think this is actually the true value of the study – that the authors are looking at continuous speech rather than single utterances. As it is currently written, it looks like the paper is framing the main contribution as the combination of EEG and pupillometry, which is not novel. I would suggest emphasizing that the main contribution is the transition from single utterance to continuous speech.

Abstract: “The effects of SNR on listening effort were explored objectively using pupillometry and EEG data.”

Omit the word “data” because the exploration was done using the method, and the data resulted from the method; the data are not the method.

Line 29: “interpreting” – this should be “perceiving”, as “interpreting” could imply that what you’re talking about is the translation of spoken language to sign language.

Line 53: “To assess listening effort, it is necessary to measure it objectively by monitoring the changes that occur in the central and autonomic nervous systems during speech processing”

This is an opinion, and it is generally though to be false; one can also measure listening effort using measures other than physiological methods. For example, people can measure reaction time, dual-task cost, subjective measures, etc. It is not *necessary* to objectively monitor autonomic changes.

Line 193: “The bias filter in this method for denoising was chosen as the average of trials”

I do not know what this sentence means. Please explain.

Line 288: “The relative decrease of MPD measurement over 30 seconds might emphasize the sustained attention of the task at hand, which, as a result, led to listening effort”

I don’t understand the logic of this sentence. I don’t see how the decreased pupil size can “emphasize” sustained attention, and I do not understand what the authors are saying led to listening effort. Please explain.

Since the stimulus/task is so prolonged compared to previous pupillometry studies, it’s not clear to me why the authors didn’t devise a new metric that is actually suited to this design. It looks like measurements suitable for short-stimulus design were used here without much reflection on whether they were appropriate. As the use of continuous signals is rather novel for this study, it is a missed opportunity for the authors to offer a new kind of analysis that is actually designed for this procedure. Slope measurements seem like a reasonable start?

Line 335: “alignment” is a confusing word here, since it implies a correlation that is later said to be absent. The authors of that work instead used the word “coregistration”, conveying the idea that the two measurements were made at the same time but does not promise a correlation.

Line 343: This section is good, since it reminds the reader that lack of correlation between EEG and pupil measures is not necessarily a problem, but perhaps a sign that they index different brain functions. However, there is a flaw in the argumentation. In lien 343: “Pupil diameter is suggested to reflect locus coeruleus–noradrenergic (LC-NE) neuro-modulatory system which increases task relevant neuronal gain in cerebral cortex (Murphy et al., 2014).” While it is not controversial to connect pupil dilation to the LC, it has also been established that pupil size correlates with a broad range of distributed cortical activity (Reimer et al., 2016). In other words, just because pupil size correlates with LC activity, that doesn’t mean it correlates *specifically* with LC activity.

Line 351: “The theta band, which oscillates in slower frequencies than alpha band, has been widely recognized as “cognitive effort” in many non-auditory working memory tasks”

I think there’s a word or two missing here – the theta band itself isn’t recognized AS effort – perhaps it is recognized as reflecting effort or indicating effort?

Acknowledgments - One Polish character is used properly in “Skłodowska” but other Polish letters were missing from “Książek”. This is nitpicky, but since it involves a person’s name I thought it would be worth correcting.

Figure 4 – this figure is difficult to read and also lacks regression lines among data subsets (i.e. between all the 0-5 timepoints, between all the 5-10 timepoints, etc.) in addition to the overall regression line. Perhaps there is a positive correlation between 10 and 15 seconds?

The color scale in Figure 4 is also quite unhelpful because there is no orderly manner to view the differences across time. If the color saturation increased / got darker with increasing time, then it would be easy to follow across the different time points. But as it’s drawn now, it’s not easy to see why the different timepoints are even shown, because the reader can’t possibly separate them or follow them along.

Reviewer #2: The authors investigated listening effort of individuals with hearing loss using both pupillometry and electroencephalography (EEG). The main findings are that the overall parietal alpha power obtained in EEG and the mean pupillometry diameter dilation both vary with target and masker ratio (referred to as SNR in this manuscript), suggesting that both measures are sensitive to listening effort, consistent with how other authors interpret these findings in the literature.

The study is generally well conceived, described and executed (albeit with a relatively small N, but provided the necessary caveats). I only have one major suggestion and some minor editorial points.

Major suggestion:

Since this is a spatial attention task, the parietal alpha power might be modulated differently across hemispheres depending on the side of attention (e.g., Deng, Reinhart, Choi and Shine-Cunningham, eLife 2019). I suggest the authors reanalyze the parietal alpha power by binning the left target and the right target speaker separately. This may provide more signal to the study.

Minor suggestions:

Ln 58: change "to measure electoral activity caused by neural oscillations" to "measure neural oscillations"

Ln 154: the reader might want to know what type of questions are being asked and what type of answers are provided in the 3-AFC behavioral measurement. Perhaps you can provide a typical question (translated) so that the readers get a sense of the scope of the question (and the level of cognitive process that might involve in getting them right)?

Ln 255: change "showed less activation" to "was decreased"

Ln 257: change "topomap" to "topographical map"

6. PLOS authors have the option to publish the peer review history of their article (what does this mean?). If published, this will include your full peer review and any attached files.

Reviewer #1: Yes: Matthew Winn

Reviewer #2: Yes: Adrian KC Lee

---

## [Author Response · Author response to Decision Letter 0]

2 Jun 2020

We have provided a PDF file in the attachments (14 pages) to thoroughly address all the concerns and comments by the reviewers.

---

## [Decision Letter · Decision Letter 1]

23 Jun 2020

An exploratory Study of EEG Alpha Oscillation and Pupil Dilation in Hearing-Aid Users During Effortful listening to Continuous Speech

PONE-D-20-04937R1

Dear Dr. Tirdad Seifi Ala

We’re pleased to inform you that your manuscript has been judged scientifically suitable for publication and will be formally accepted for publication once it meets all outstanding technical requirements.

Kind regards,

Ifat Yasin

Academic Editor

PLOS ONE

Additional Editor Comments (optional):

Reviewers' comments:

Reviewer's Responses to Questions

**Comments to the Author**

1. If the authors have adequately addressed your comments raised in a previous round of review and you feel that this manuscript is now acceptable for publication, you may indicate that here to bypass the “Comments to the Author” section, enter your conflict of interest statement in the “Confidential to Editor” section, and submit your "Accept" recommendation.

Reviewer #1: All comments have been addressed

Reviewer #2: All comments have been addressed

2. Is the manuscript technically sound, and do the data support the conclusions?

Reviewer #1: Yes

Reviewer #2: Yes

3. Has the statistical analysis been performed appropriately and rigorously? 

Reviewer #1: Yes

Reviewer #2: Yes

4. Have the authors made all data underlying the findings in their manuscript fully available?

Reviewer #1: No

Reviewer #2: No

5. Is the manuscript presented in an intelligible fashion and written in standard English?

Reviewer #1: Yes

Reviewer #2: Yes

6. Review Comments to the Author

Reviewer #1: All of my concerns from the first submission have been adequately addressed. The language is smoother and the additional introductory material really strengthen the paper.

Reviewer #2: The authors have adequately addressed concerns raised by previous reviewers. I have no further comments.

7. PLOS authors have the option to publish the peer review history of their article (what does this mean?). If published, this will include your full peer review and any attached files.

Reviewer #1: Yes: Matthew B. Winn

Reviewer #2: Yes: Adrian KC Lee

---

## [Editor Report · Acceptance letter]

30 Jun 2020

PONE-D-20-04937R1 

An exploratory Study of EEG Alpha Oscillation and Pupil Dilation in Hearing-Aid Users During Effortful listening to Continuous Speech 

Dear Dr. Seifi Ala:

I'm pleased to inform you that your manuscript has been deemed suitable for publication in PLOS ONE. Congratulations! Your manuscript is now with our production department. 

Kind regards, 

on behalf of

Dr. Ifat Yasin 

Academic Editor

PLOS ONE